# Pathogenic Roles of S100A8 and S100A9 Proteins in Acute Myeloid and Lymphoid Leukemia: Clinical and Therapeutic Impacts

**DOI:** 10.3390/molecules26051323

**Published:** 2021-03-02

**Authors:** Julie Mondet, Simon Chevalier, Pascal Mossuz

**Affiliations:** 1Department of Biological Hematology, Grenoble Alpes University Hospital, 38400 Grenoble, France; jmondet@chu-grenoble.fr (J.M.); schevalier@chu-grenoble.fr (S.C.); 2Institut for Advanced Biosciences, 38700 La tronche, France; 3UF de Pathologie Moléculaire, Département d’Anatomie et Cytologie Pathologiques, CHU Grenoble, 38043 Grenobl, France; 4Laboratoire d’Hématologie Cellulaire, Institut de Biologie et Pathologie, CHU Grenoble, 38043 Grenoble, France

**Keywords:** S100A8/S100A9, acute myeloid leukemia, acute lymphoid leukemia, inflammation in cancer

## Abstract

Deregulations of the expression of the S100A8 and S100A9 genes and/or proteins, as well as changes in their plasma levels or their levels of secretion in the bone marrow microenvironment, are frequently observed in acute myeloblastic leukemias (AML) and acute lymphoblastic leukemias (ALL). These deregulations impact the prognosis of patients through various mechanisms of cellular or extracellular regulation of the viability of leukemic cells. In particular, S100A8 and S100A9 in monomeric, homodimeric, or heterodimeric forms are able to modulate the survival and the sensitivity to chemotherapy of leukemic clones through their action on the regulation of intracellular calcium, on oxidative stress, on the activation of apoptosis, and thanks to their implications, on cell death regulation by autophagy and pyroptosis. Moreover, biologic effects of S100A8/9 via both TLR4 and RAGE on hematopoietic stem cells contribute to the selection and expansion of leukemic clones by excretion of proinflammatory cytokines and/or immune regulation. Hence, the therapeutic targeting of S100A8 and S100A9 appears to be a promising way to improve treatment efficiency in acute leukemias.

## 1. Introduction

Human proteins S100A8 and S100A9 are low molecular weight proteins (around 10kDa) formed by 93 and 114 amino acids, respectively. Also known as “myeloid-related protein” (MRP-8/MRP-9) and calgranulin (A/B) respectively, they belong to the S100 protein family, which has about 20 members [1,2]. The genes encoding the two proteins S100A8 and A9 are located on locus 1q21. The members of this family contain a common structure consisting of two “EF-hand” motifs, themselves made up of two alpha chains separated by a peptide loop in which the ion Ca^2+^ can bind. Between the two “EF-hand” motifs, there is a hinge region variable between S100 proteins that most often confers the biological and specific activity of different S100 proteins. 

Like all proteins in the S100 family, the S100A8 and S100A9 could form monomers, homodimers, or a heterodimer S100A8/S100A9, also called calprotectin. They are expressed constitutively in the cytoplasm of myeloid cells, including myeloid precursors, polynuclear neutrophils (PNN), and monocytes, but are absent from lymphocytes [3,4,5]. They appear at the promyelocytic stage to become maximum in mature granulocytes where they account for up to 40% of total cytosolic proteins [6]. In monocytes, they are present in the mature stages of the monocytic lineage, representing 1 to 5% of monocytic proteins, while they used to disappear when monocytes differentiate into macrophages under physiological conditions. 

Thanks to their structure with two “EF-hands”, S100A8 and A9 proteins are considered calcium biosensors changing their conformation in response to calcium influx and thus, their affinity to proteins. In particular, the increase in intra-cellular calcium of granulocytes causes the link of hetero-dimer S100A8/A9 with arachidonic acid, which would be released into the extracellular space, allowing the synthesis of precursors of inflammation such as leucotrienes or prostaglandins. One of the other major roles of the S100A8 and A9 is the activation of the complex of NADPH oxidase (NOX), which allows the destruction of pathogens through the production of reactive oxygen species (ROS) [7,8]. The heterodimer S100A8/A9 is also involved in the polymerization of tubulin and thus in the rearrangement of the cytoskeleton that is mandatory for the diapedesis and migration of granulocytes [9].

Like many members of the S100 protein family, S100A8 and A9 can be secreted in extra-cellular space via an active calcium-dependent mechanism of translocation from the cytoplasm to cellular membrane [10,11,12]. In the extracellular space, both S100A8 and S100A9, either as monomers, homo-dimers or hetero-dimers can interact with receptors of the cytoplasmic membrane. Toll-like receptors 4 (TLR4) and receptor for advanced glycation end products (RAGE) have been frequently involved in transducing dependent S100A8/S100A9 signals [13,14,15]. In more detail, S100A9 mainly signals via TLR4 and can also associate with CD147/basigin [16]. RAGE is described as the main receptor for S100A8; however, the relevance of the TLR4 signaling pathway for S100A8 was experimentally validated using human and murine models of TLR4 [17]. Other receptors of the S100A8 or hetero-oligomer S100A8/A9 protein have been described, such as differentiation Cluster (CD36), CD11b/CD18 (Mac-1), CD69, CD33, MCAM receptor, neuroplastin-β or PAR-2, as well as the ability to interact with membrane proteoglycan [3,18,19,20,21].

Overall, the S100A8 and A9 proteins are considered as alarmins, otherwise known as damage-associated molecular pattern (DAMP). Under inflammatory conditions, the protein expression of S100A8 and S100A9 is induced in a large variety of cells such as endothelial cells, macrophages, osteoclasts, keratinocytes and dendritic cells [22]. Thus, elevated serum levels of S100A8 and/or S100A9 are found in many inflammatory pathologies (psoriasis, rheumatoid arthritis, chronic inflammatory intestine diseases, etc.). S100A8 and S100A9 could also amplify the inflammatory response by secretion of pro-inflammatory cytokines (TNF, IL6, etc.) via its link to the TLR4 receptor and the induction of the NF-kB and Mitogen-activated protein kinases (MAPK) pathway [23]. They also participate in the ROS-dependent activation of the NLRP3 inflammasome [24]. In addition to their activity in cytokines secretion, they exert a chemo-attractive function allowing the recruitment and leukocyte adhesion [3,17,24,25]. The process of cell adhesion is mediated by the fixation of S100A8 and/or A9 on the neutrophils TLR4 that improves the expression of Intercellular Adhesion Molecule 1 (ICAM-1) and Vascular cell adhesion protein 1 (VCAM) adhesion molecules by endothelial cells allowing leukocytes to adhere to the endothelial wall.

The major impact of the inflammatory reaction in carcinogenesis largely explains the role of S100A8 and A9 in the progression and development of tumors. There are ample evidences in the literature to support the pro-tumoral impact of S100A8 and A9 in many solid cancers. Firstly, gain or amplification of 1q21 locus of S100 genes are common in cancer (Figure 1) from [26,27]. Moreover, S100A8 and S100A9 proteins contribute to tumor growth, metastasis development, angiogenesis, and immune escape in a wide variety of solid cancers [2,28,29]. Indeed, the expression of proteins S100A8 and S100A9 is up-regulated in several tumor types (breast, prostate, colon, melanoma, etc.). S100A8 have been shown to be overexpressed in aggressive phenotype [28] and in patients with low overall survival [29], suggesting that regulation of S100A8 gene expression is a mechanism that participates in cancer cells control. Proteins S100A8, S100A9, or hetero-oligomer S100A8/A9 might also promote cancer cells migration and dissemination through regulation of tumoral microenvironment [30]. Similarly, there are increasing data about the pathological role S100A8 and S100A9 play in the development of hematological malignancies and in particular in acute leukemias (AL).

In this article, we present a review of data from the literature concerning the deregulation of the expression of S100A8 and S100A9 in acute myeloid and lymphoid leukemia (AML and ALL, respectively), along with an analysis of their impacts on the proliferation and apoptosis of leukemic cells that integrates the action of both proteins on the hematopoietic microenvironment. The consequences, in terms of chemoresistance and the modification of the patient’s prognosis are presented. Finally, a synthesis of the different therapeutic strategies targeting S100A8 and S100A9 and their potential in the treatment of acute leukemia is discussed.

## 2. Pathogenic Role of S100A8 and S100A9 in Acute Myeloid and Lymphoid Leukemia

### 2.1. Gene and Protein Expression of S100A8 and S100A9 in Leukemic Cells

Numerous data from the literature suggest that there is a deregulation of the expression of coding genes for S100A8 and/or S100A9 in acute myeloid leukemia (AML). The "genome landscape" study showed by RNA sequencing on a cohort of 200 newly diagnosed AML patients that a significant proportion of these patients, mainly M4 and M5 subtypes of the French American British (FAB) classification, over-expressed transcripts of S100A8 and S100A9 [31]. These results were supported by Laouedj et al., on 437 AML (excluding acute promyelocytic leukemia (APL)) that showed an overexpression of the S100A8 and A9 transcripts in the AML M4 and M5 compared to the poorly differentiated AML (AML M0 and M1) [32].

A more recent study of 189 AML patients (without APL), including a non-AML patient control group, showed that at diagnosis, the average level of expression of S100A8 and S100A9 mRNA was in fact lower in AML patients than those in the control group [33]. Moreover, they showed that relapsed patients were lower than patients in complete remission who harbored lower but not significantly different rates from the control group. This is likely due to the fact that the bone marrows of controls and patients in complete remission are enriched (compared to AML at diagnosis or in relapse) in mature cells of the granular lineage that contain the highest concentration of S100A8 and A9. Thus, to understand the clinical value of S100A8 and A9, it seems more informative to study and compare their variations within different cytological, prognostic, or molecular subtypes of AML rather than in relation to a control group of non-AML patients.

Indeed, one common feature of all comparative studies is the variability of expression levels between the different subtypes of AML, that commonly showed a relative over-expression in monocytic M4 and M5 AML in comparison to the M1 and M2 subtypes [5,32,33]. This raises the delicate question of the relative proportion of S100A8 and A9 originating from blast cells (promonocyte, monoblast and myeloblast) and/or secreted by monocytes associated with the reactive inflammatory reaction in monocytic AML. A more recent study using micro-array showed that overall AML M1 and M2 patients had similar or slightly lower levels than healthy subjects [34]. On the other hand, further analysis shows that AML M2 patients (more differentiated AML) have higher levels than AML M1 that were relatively close to healthy subjects.

However, very few data are available about a potential link between S100A8 and S100A9 expression and genetic characteristic of AML. A study of *IDH* (Isocitrate deshydrogenase) mutated pediatric AML showed that the two most downregulated genes in *IDH* mutated children were S100A8 and S100A9 genes, suggesting a link between *IDH* mutation and S100 genes expression. Moreover, *IDH* mutations in pediatric AML were associated with a better prognosis compared to wild type *IDH*, indirectly suggesting that S100A8 and S100A9 also play a prognosis role in pediatric AML [35].

In acute promyelocytic leukemia (APL), the FAB M3 AML subtype, S100A9 and A8 seem to be expressed less compared to other subtypes despite the promyelocytic phenotype of these blasts. Jalili et al. showed that APL expressed the lowest gene transcripts among all FAB classes [36]. Moreover, S100A9 levels was showed to be inversely correlated with the expression levels of PML/RARA, the fusion gene, which defines APL. In an in-silico analysis of public databases, TCGA-LAML (n = 150 by RNAseq) and NCBI GEO GSE37642 (n = 136 by microarrays), we found that these differences vary according to the method for the evaluation of gene expression. S100A8 transcript levels evaluated by RNAseq in APL were lower than other FAB subtypes (no RNA seq data for S100A9), but with microarray, the expressions of both S100A8 and S100A9’s were lower but not significantly different in APL compared to other FAB subtypes (data not published).

In pediatric acute lymphoblastic leukemia (ALL), as expected, since normal lymphoid cells do not express S100A8 and S100A9, the expression levels of these genes were significantly lower than in AML. However, a meta-analysis of public data from 3 studies showed that S100A8 is the top upregulated gene in relapse ALL compared to the diagnosis [37]. In addition, a significant increase in the expression of the S100A8 gene was found in the most aggressive forms of B-ALL, as well as in *Mixed Lineage Leukemia (MLL)* rearranged ALL resistant to prednisolone [38]. The markedly enhanced expression of S100A8 and S100A9 in infant ALL suggests that S100 proteins may play an important role in leukemogenesis and/or adverse clinical course of ALL during infancy.

Proteomics study first showed that the expression of the S100A8 and S100A9 proteins was 9 to 14 times lower in AML than in granulocytes of healthy subjects [39]. Within the FAB subtypes, the highest levels were found in AML M4 and M5, but also in APLs with expression rates five times higher in APL and AML M2 compared to AML M1, which was expected due to the granular differentiation of these AML, but was quite different compared to gene expression. As expected, the S100 expression in AML was significantly stronger than in ALL. In ALL, another study showed that S100A8 and S100A9 protein expression decrease in resistant patients compared to the complete remission group. However prolonged follow-up shows that patients in prolonged remission have lower rates than patients in short remission, and that patients resistant to chemotherapy or dead, plead for an adverse role of S100 proteins [40]. In that sense, in a MS based-proteomics study, we showed in 54 patients that the S100A8 protein was significantly overexpressed in patients with an unfavorable prognosis. In addition, analysis of S100A8 expression in the intermediate cytogenetic group allow to identify two different prognostic subgroups based on the value of S100A8 [41].

### 2.2. Secretion in Blood and Bone Marrow Plasma of S100A8 and S100A9 Proteins

Studies that assessed blood plasma/serum levels or bone marrow plasma secretion of S100A8 and S100 A9 or heterodimer A8/A9 have brought additional informative data. Laouedj et al. reported that AML patients displayed significantly higher levels of S100A8 in plasma levels, and using a mouse model of AML, they argue that this secretion came from leukemic cells and not microenvironment [32]. Mondet et al. showed in a study performed on 78 bone marrow (BM) plasmas including AML, myeloproliferative neoplasms (MPN), myelodysplastic syndromes (MDS) and healthy donors (HD), that AML patients displayed significantly higher S100A8 levels than MPN, MDS and HD [5]. Among molecular subgroups, a higher concentration of S100A8 in bone marrow plasma was observed in AML with *FLT3-ITD* mutation, in NPM1 muted AML or with a tendency in inv(16) (p13.1q22).However, intracellular measurement of S100A8, by flow cytometry highlighted that it mainly originated from both monocytes and leukemic cells expressing monocytic markers such as promonocytes and monoblast, arguing for the role of bone marrow microenvironment in secretion of S100 proteins. Finally, they reported that high levels of S100A8 could be associated with a worst overall survival in M4/M5 subgroups but not in global population. 

Another study that focused on circulating S100A8 and S100A9 homodimers and S100A8/A9 heterodimers in plasma from 58 severe aplastic anemia (SAA) and 30 MDS patients, showed that circulating S100A8 was increased in MDS compared to those of SAA and/or healthy controls, and that it may be useful to distinguish these diseases in the differential diagnosis of bone marrow failure syndromes. However, the highest level of S10018 were observed in low risk MDS, suggesting that the secretion of S100A8 is not a marker of worse prognostic in MDS. Interestingly, these authors also highlighted that differences exist between therapy responder or not responder according to their profile of secretion. Non-responding patients had significantly higher levels of circulating S100A8/A9 heterodimers compared to responders and healthy controls, but without variations of S100A8 and S100A9 homodimers [42].

### 2.3. Role of S100A8 and S100A9 on Proliferation and Apoptosis of Leukemic Cells 

The roles of S100A8 and S100A9 have been studied in many tumor models (colon, breast, neuroblastoma, etc.) and seems to be dual. S100A8/A9 heterodimer promotes tumor cell growth at low concentrations through RAGE signaling and NF-κB-dependent phosphorylation of p38 and p44/42 MAPKs in MCF-7 and MDA-MB231 cells. Conversely, high doses (10 μg/mL) of recombinant human S100A8/A9 suppressed the growth of MC38adenocarcinomacell line [15,43].

As observed in models of solid tumors, S100A8 and S100A9 are dose-dependent regulators of myeloid differentiation and leukemic cell proliferation, low doses being pro-tumoral and higher doses exerting a negative regulatory role on leukemic cells proliferation [44]. In fact, in our team, we showed (Figure 2; unpublished data) that leukemic cells from acute myeloid leukemia cell lines display a great variability of response to rhS100A8, depending on their phenotype and genotype. In the presence of rhS100A8, HL60 exhibited a dose-dependent proliferation associated with a dose-dependent decrease of ROS emission. On the other hand, rhS100A8 induced in OCI-AML3 cell line a dose-dependent apoptosis associated with an increase of ROS production, while KG1 cell line were almost insensitive to the addition of rhS100A8. In total, these experimental data suggest that the effects of S100A8 on leukemic cells probably depend on their genetic variability, but also on the expression of the many surface receptors that could transmit its activity (RAGE, TLR4, CD36, CD11b/CD18 (Mac-1), CD69, etc.).

Moreover, it seems that S100A8 and S100A9 could play different or even contradictory roles, respectively. In vitro and in vivo mice models showed that S100A9 induced AML cells differentiation whereas S100A8 blocked it. Investigating the pathways involved in S100A9-induced AML cell differentiation revealed that S100A9 induces differentiation via TLR4 and several downstream factors, including p38 MAPK, extracellular signal-regulated kinases (ERK1/2), Jun N-terminal kinase (JNK), and NF-κB. Moreover, treatment with an anti-S100A8 antibody induced a similar impact on AML cell differentiation as observed with treatment with recombinant S100A9 protein that both prolonged survival in the same mouse model of AML [32]. These data suggest an antagonistic relationship between S100A8 and S100A9 and that the relative ratio between S100A9 and S100A8 determines the degree of their impact on AML differentiation. It is noteworthy that these opposite roles of S100A8 and S100A9 proteins were described on AML M4 and M5 of FAB classification, and that it should be very fruitful to make comparisons in other AML subtypes.

In addition, S100A8 and S100A9 proteins could mediate cell death through several mechanisms: first, by caspase 3-dependant apoptosis [45] induced by depletion in Zn^2+^, and the reduction of mitochondrial membrane potential, causing the release of Smac/Diablo and Omi/HtrA2 (without cytochrome C) that disrupt the balance between pro and antiapoptotic proteins [46]. In addition, S100A8 and S100A9 have shown to induce cell death by autophagy. The treatment of cells with S100A8/A9 causes the translocation of BNIP3, a member of the Bcl2 family, to mitochondria and induces an increase in ROS, subsequently followed by mitochondrial damage and lysosomal activation, leading to cell death via autophagy [47]. This S100A8/A9-induced autophagy appears to be independent on the RAGE receptor. Additionally, S100A8 and S100A9, which bind to TLR4 and CD33, can stimulate pyroptosis by enhancing the production of inflammasome component leading to the secretion of proinflammatory cytokines such as pro-IL-1b and pro-IL-18. Indeed, S100A9 could mediate the activation of NOX that generates ROS, contributing to inflammasome assembly and inflammatory cytokine production, β-catenin activation, and thereby trigger pyroptosis in MDS bone marrow mononuclear cells (BM-MNCs) [48]. Reciprocally, inhibition of S100A9 reduced transcriptional priming of pyroptosis-associated genes (CASP1, IL-1B, IL-18, and NLRP3) and improved the colony forming capacity of BM-MNCs from patients with MDS. These findings were validated in a cohort study on133 patients and 31 healthy controls which showed that apoptosis associated protein that recruits caspase (ASC) percentage correlated with S100A8 and S100A9 bone marrow plasma levels [49]. In a similar way, it was shown that S100A8 and S100A9 both induce apoptosis of chronic eosinophilic leukemia (CEL) cells through impact on signaling and mRNA expression of FIP1L1-PDGFRa. Moreover, both S100A8 and A9 proteins could inhibit tumor progression of xenograft model of CEL [50].

Finally, S100A8 and S100A9 also seem to regulate leukemic proliferation, through impact on immune check point. Programmed death 1(PD-1) and programmed death-ligand 1 (PD-L1) can be activated in S100A9 transgenic (S100A9Tg) mice and by treatment of BM-MNC with S100A9. Further, MDS BM-MNC treated with recombinant PD-L1 underwent cell death, and reciprocally, PD-1/PD-L1 blockade restores effective hematopoiesis and improves colony forming capacity in BM-MNC from MDS patients. Similar findings were observed in aged S100A9Tg mice, considering that c-Myc is required for S100A9- induced upregulation of PD-1/PD-L1 [51].

### 2.4. Impact of S100A8 and S100A9 on Bone Marrow Micro-Environment

One important impact of S100A8 and S100A9 proteins on leukemic cells is also mediated through their action on the recruitment of myeloid derived suppressor cells (MDSC), which promote tumor progression and establish a favorable pro-tumoral niche. Burke et al. show that exosomes derived from MDSC contained hetero-oligomer S100A8/A9, which could induce a RAGE dependent autocrine loop that promotes the accumulation of MDSCs within the tumor niche [52], and in fine triggered hematopoietic stem cells and progenitor cells death, contributing to ineffective hematopoiesis. Similarly, De Veirman et al. showed that S100A9 acted as a chemotractant for multiple myeloma cells and promoted secretion of inflammatory and pro-myeloma cytokines (TNFa, IL6 and IL10) that could represent a way of reducing tumor development [53]. In another study, Chen et al. showed that the plasma concentration of S100A9 significantly increased in MDS patients and that, in a murine model of S100A9 expression, S100A9 drived expansion and activation of MDSC that contributed to cytopenia and myelodysplasia. This may include engagement of other cognate receptors of the protein, including expansion of MDSCs through CD33 signaling [54].

This impact of S100A8 and S100A9 proteins on hematopoietic microenvironment has been highlighted by a transcriptional analysis of mesenchymal cells CD271+ of healthy donors compared to different pre-leukemic models (SBDS −/− mouse; patients low risk MDS patients Blackfan-Diamond anemia) [55]. The authors showed that the genes of S100A8 and S100A9 were involved in the genotoxic stress involved in the leukemia expansion process via activation of the TLR4-S100A8/A9-p53 signaling pathway. S100A8/S100A9 expression was associated with activated p53 and TLR4 signaling in mesenchymal cells and it predicted leukemic evolution in a cohort of homogeneously treated low-risk MDS patients. Finally, it was shown in the mouse model of Rps14 inactivation that mutant erythroblast expressed high levels of S100A8 and S100A9 leading to a p53-defect of erythroid differentiation that could be reset by genetic inactivation of S100A8 [56].

Hence, both S100A8 and S100A9 can contribute to the selection and expansion of leukemic clones via TLR4 and RAGE expression on normal and leukemic stem cells, together with modulation of the excretion of proinflammatory cytokines and/or immune response in micro-environment (Figure 3). It is therefore a possibility that both proteins S100A8 and S100A9 might be of particular relevance to hematological malignancies through their critical role in the progression of pre-leukemic states as myelodysplasic syndrome or myeloproliferative neoplasm. Moreover, the particular prevalence of clonal hematopoiesis of indeterminate potential (CHIP) in elder patients, who suffer from chronic inflammatory diseases very frequently, raise the hypothesis that inflammation-associated secretion of S100A8 and S100A9 could be a factor that participates in the evolution of CHIP toward frank acute myeloid leukemia, which frequency significantly increases with age.

### 2.5. Impact on Chemoresistance of Leukemic Cells of S100A8 and S100A9

Due to its multiple impacts on leukemic cells viability and apoptosis, it is not surprising that S100A8 and S100A9 can modulate chemosensitivity of leukemic cells and thereby influence the prognosis of patients. High levels of S100A8 and/or S100A9 transcripts are frequently associated with a worse prognostic and a lower survival in AML and ALL, (see Section 1). Similarly, overexpression of S100A8 protein by leukemic cells has been shown to correlate significantly with a worse prognosis [41] and S100A8 levels in bone marrow plasma could impact survival in M4 and M5 AML [5].

First, S100A8 and S100A9 have been shown to impede the action of inhibitor of apoptosis. Karjalainen et al. detected negative associations between S100A8 and S100A9 levels and sensitivity to venetoclax, a B Cell-Lymphoma (BCL-2) inhibitor [57]. AML cell lines overexpressing these proteins released less Ca^2+^ after venetoclax treatment, suggesting that the S100A8/A9 complex may mediate resistance in AML by hindering the migration of Ca^2+^ toward the mitochondrion and subsequently inhibiting apoptosis. In addition, knockdown of S100A8 expression increased the sensitivity of leukemia cells to chemotherapy and apoptosis. Similarly, inhibiting the expression of S100A8 can reverse chronic myeloid leukemia cell resistance to doxorubicin, and this may be explained by the increased intracellular Ca^2+^ levels and apoptosis induced by endoplasmic reticulum stress [58].

Second, Cao et al. demonstrated that S100A8 regulates autophagy in response to arsenic trioxide or vincristine treatment by competitively binding to the autophagy regulator beclin-1 (BECN1) [59]. S100A8 leads to a dissociation of Beclin 1 (BECN1)–BCL-2 complex, which activates autophagy and inhibits apoptosis, inducing arsenic trioxide and vincristine resistance in AML cell lines. Conversely, suppressing S100A8 expression decreased autophagy as evaluated by the formation of autophagosomes. In *another in vitro* study, S100A8 overexpression in HL-60 leukemia cell line enhanced resistance to etoposide, whereas BCL-2–associated X protein (BAX) and caspase-3 were downregulated at both the mRNA and protein levels, suggesting that S100A8 promotes AML chemotherapy resistance by downregulating the mitochondrial apoptosis pathway [33]. Consistent with these findings, S100A8 and S100A9 have shown to be upregulated in AML cells after daunorubicin chemotherapy, thereby promoting chemoresistance [60].

In APL, upon treatment with all-trans retinoic acid (ATRA) and, to a lesser degree, with As2O3, S100A9 expression levels increase but also seem to have increased sensitivity to treatment through a pro-apoptotic impact. These results were confirmed in the APL cell line NB4, where higher levels of S100A9 induced apoptosis through reduced Bcl-2 levels, cleavage of caspase 3, and leukemic cell growth suppression. Furthermore, higher S100A9 mRNA levels correlated with higher expression of PU.1, an important transcription factor for myeloid differentiation. Thus, at least in APL, higher S100A9 levels are linked with myeloid differentiation, leukemia growth suppression, and increased treatment response [44,61].

The markedly enhanced expression of S100A8 and S100A9 in poor prognostic infant ALL suggests that these S100 proteins may also play an important role in chemosensitivity of ALL during infancy. Indeed, increased co-expression of S100A8 and S100A9 proteins synergistically promote cell survival by exhibiting marked anti-apoptotic activity [62]. Increased expression of S100A8 and S100A9 have been also associated with steroid resistance [63] and relapse in childhood ALL [64]. In primary prednisolone-resistant *MLL*-rearranged infant ALL, S100A8, and S100A9 mRNA levels appear extremely high. Over-expression of S100A8 and S100A9 is associated with failure to induce free-cytosolic Ca^2+^ together with prednisolone resistance. Finally, enforced expression of S100A8/S100A9 in prednisolone-sensitive *MLL*-rearranged ALL cells, rapidly leading to prednisolone resistance as a result of S100A8/S100A9 mediated suppression of prednisolone-induced free-cytosolic Ca^2+^ levels. Another hypothesis is that S100A8/S100A9 may exert direct inhibitory effects by effectively blocking Ca^2+^ release at its source without the necessity of binding high concentrations of free-cytosolic Ca^2+^ [65,66].

## 3. Landscape of S100A8 and S100A9 Therapeutic Strategies

As S100A8 and A9 appear as key regulators of leukemic cells survival and participate in mechanisms of chemoresistance, targeting of these S100 proteins represents a tempting and promising way for new therapeutic. First, it could be noticed that, in murine models of breast cancer, S100A9 antibodies have been used in conjunction with single-photon emission computed tomography (SPECT) for the in vivo detection of S100A8/A9 as a marker for the establishment of the pre-metastatic niche suggesting the interest of evaluating S100A8 and S100A9 for the stratification of patients [67,68]. Similarly, numerous studies that showed an increase of gene expression and/or protein levels in bad prognosis subgroups (see Section 2.1) argue for the interest of evaluation at genetic and protein levels of S100A8 and S100A9 in the patient’s stratification at diagnosis. From a therapeutic point of view, different strategies targeting S100A8 and/or S100A9 directly or indirectly have been evaluated in AML and ALL (Table 1).

S100 neutralizing antibodies represent a promising way to block action of S100 in cancers. To date, antibodies targeting S100A8/A9, S100A4, S100A7 [69] and S100P have demonstrated efficacy for a number of pathological conditions [70]. In addition, peptibodies, peptide-Fc fusion proteins that target S100A8 and S100A9, reduce tumor burden in multiple cancer models [71]. Moreover, S100A8 and S100A9 neutralizing antibodies have been shown to block the recruitment of both myeloid cells and circulating tumor cells [72,73]. In mice models of acute myeloid leukemia (AML), S100A8 antibodies, but not S100A9 antibodies, induce AML cell differentiation, decrease leukemic burden and increase survival [32].

The knowledge about glucocorticoids resistance in *MLL*-rearranged infant ALL has paved the way to new therapeutic options. The Src (Sarcoma) kinase inhibitor PP2 markedly sensitized *MLL*-rearranged ALL cells otherwise resistant to prednisolone, via downregulation of S100A8 and S100A9, which allowed prednisolone-induced Ca^2+^ fluxes and trigger apoptosis [66]. However, as this compound was designed to target a broad spectrum of Src kinase family members, one cannot exclude that their impact on prednisolone resistance may be mediated through additional action on Src kinase inhibition. Also, S100A8 and S100A9 may not be the only S100 family members targeted by this inhibitor, as PP2 was recently shown to also inhibit S100A4 expression in colon carcinoma. Hence, small molecules inhibitors that prevent the formation of the S100A8/S100A9 heterodimeric complex, or that block the Ca^2+^ binding sites in these proteins, could be a potential way to diminish its suppressive actions on prednisolone-induced Ca^2+^ release, and allow apoptosis [66].

The link between tumor infiltrating monocytes/macrophages and S100A8 and S100A9 expression in cancer cells supports their role in diffusion and spreading of cancer cells. These findings suggest that S100A8 and S100A9 may serve as useful targets for anti-metastasis therapy. Indeed, inhibition of S100A9 using quinoline-3-carboxamide derivatives such as tasquinimod has shown anti-tumor effects in several pre-clinical models, through modulation of the tumor microenvironment. Tasquinimod inhibited myeloid-derived suppressor cell recruitment and infiltration, leading to enhanced tumor immunity and decreased angiogenesis [74]. In the (murine mouse lymphoblastic lymphoma cell line (el4)), tasquinimod, through blockade of the interaction of S100A9 with its membrane receptors TLR4, could inhibit tumor growth, which was associated with reduced expression of transforming growth factor (TGFβ) [75]. A further study indicates that suppression of S100A8 and S100A9 in cancer cells using short hairpin RNA significantly diminished migration and invasion in culture [30]. By analogy, S100A8 and S100A9 have been shown to play a critical role in the evolution of pre leukemic clones and in worsening of MDS patients from low risk to high risk. Therefore, it is not meaningless to hypothesize that preventive action on S100A8 and/or S100A9 levels or expression should help to down modulate the clonal evolution and to prevent leukemia emergence.

Finally, one interesting model of S100A8 and S100A9 targeted therapy is provided by the use of inhibitors of Bromodomain-containing protein 4 (BRD4), a member of the BET bromodomain family of transcriptional regulators. Antileukemic mechanisms induced by BET inhibitors are currently not well understood but recently, a study showed that in AML cells, S100A8 and S100A9 expression could be downregulated by JQ1, a benzodiazepine molecule and a specific BET inhibitor in some AML cell lines (i.e., OCI-AML3) [76]. Daunorobicin is known to cause a dose- and time-dependent increase in S100A8 and S100A9 that could mediate chemoresistance through autophagy. However, when JQ1 is associated with daunorubicin, JQ1 synergizes with daunorubicin to cause apoptosis in AML cell lines via suppression of S100A8 and S100A9 levels. The authors hypothesized that JQ1 stimulates apoptosis by overcoming autophagy in an S100A8/A9-dependent manner [77].

Altogether, these data suggest that strategy targeting S100A8 and/or S100A9 might be fruitful at different stages of development of acute leukemia (Figure 4). First, in the pre leukemic state, regulation of inflammatory production of both proteins could delay or impede the expansion of leukemic clone toward a secondary acute leukemia. In particular, high risk MDS patients could benefit from antibody-based targeting of S100A8/A9 proteins, but also from therapy, such as tasquinimod, which regulates the action of S100 protein in the microenvironment. For de novo acute leukemia, antibodies targeting S100A8/A9 might be proposed for worse prognosis patients as MLL re-arranged ALL or height risk AML. For refractory patients, regulation of chemosensitivity should be proposed using small molecule inhibitors against the complex A8/A9, or through transcriptional regulation by BET inhibitor.

In conclusion, a deregulation of gene and/or protein expression of S100A8 and/or S100A9 in leukemic cells of AML and ALL appears repeatedly and solidly in many studies, sometimes associated by secretion of proteins in the peripheral plasma as well as in the bone marrow microenvironment. This dual location (cellular and extracellular) explains the prognostic impact of these proteins in AML and ALL patients that result from the combination of their effects on the proliferation and survival of blast cells, but also from extra-cellular mechanisms linked to secreted S100A8 and. Hence, the therapeutic targeting of S100A8 and S100A9 appears as a promising way to improve treatment efficiency in acute leukemias. 

## Figures and Tables

**Figure 1 molecules-26-01323-f001:**
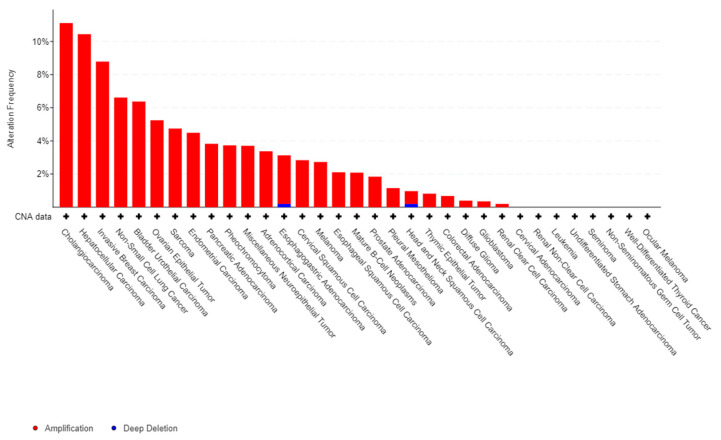
Frequency of alteration of 1q21 locus alterations in cancer. The histogramme represents the frequency (in percentage) of alteration of 1q21 locus. Data extracted from cbioportal databases [26,27].

**Figure 2 molecules-26-01323-f002:**
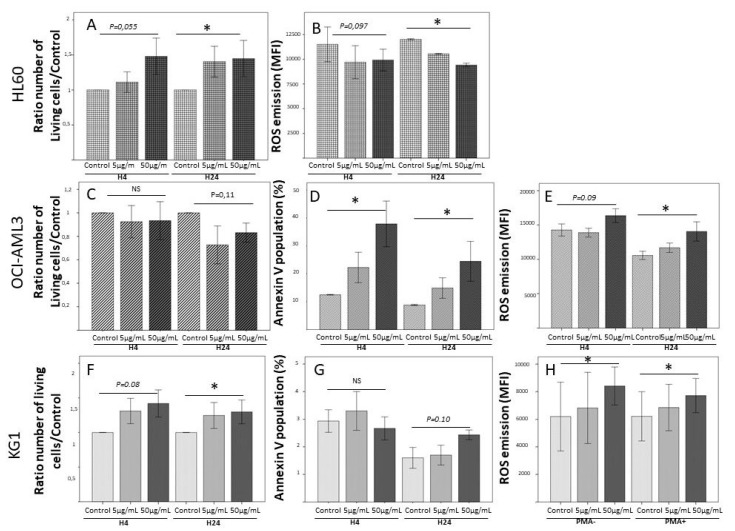
Pleiomorphic impact rhS100A8 on 3 human leukemia cell lines. Cells (HL60, OCI-AML3 and KG1) were cultured in RPMI+SVF for 24 h in the presence of two concentrations of rhS100A8 (5 µg/mL and 50 µg/mL) in comparison to culture without rhS100A8. PMA was introduced as a test of ROS emission by leukemic cells. Viability, apoptosis, and ROS emission were assessed by FACS analysis at 4 h and 24 h. HL60: (**A**) Cell growth impact after H4 and H24 of S100A8 incubation. Bar graph represents mean+-Standard deviation. (**B**) ROS emission in non-apoptotic population after H4 and H24 of S100A8 incubation. Bar graph represents mean+-Standard deviation. * means *p* < 0.05. Abbreviation: MFI Mean of Fluorescence intensity; OCI-AML3: (**C**) Cell growth impact after H4 and H24 of S100A8 incubation; (**D**) Proportion of apoptotic cells by annexin-V measure after H4 and H24 of S100A8 incubation; (**E**) ROS emission by global population after H4 and H24 of S100A8 incubation. KG1: (**F**) Cell growth impact after H4 and H24 of S100A8 incubation; (**G**) Proportion of apoptotic cells by annexin-V measure after H4 and H24 of S100A8 incubation; (**H**) ROS emission by non-apoptotic population after H24 incubation with S100A8 with or without PMA addition.

**Figure 3 molecules-26-01323-f003:**
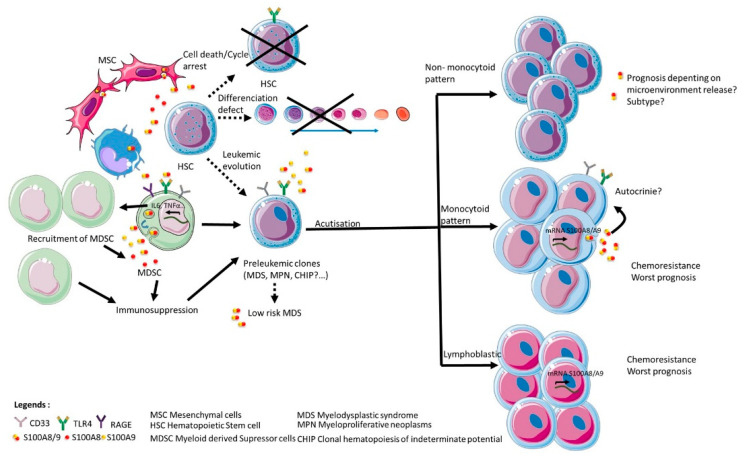
Impact of S100A8 and S100A9 in disruption of hematopoiesis and leukemic progression. The figure is a representation of the impacts described on the literature of S100A8 and/or S100A9 as homodimers or heterodimers on normal hematopoietic stem cell and on the emergence and progression of leukemic clones.

**Figure 4 molecules-26-01323-f004:**
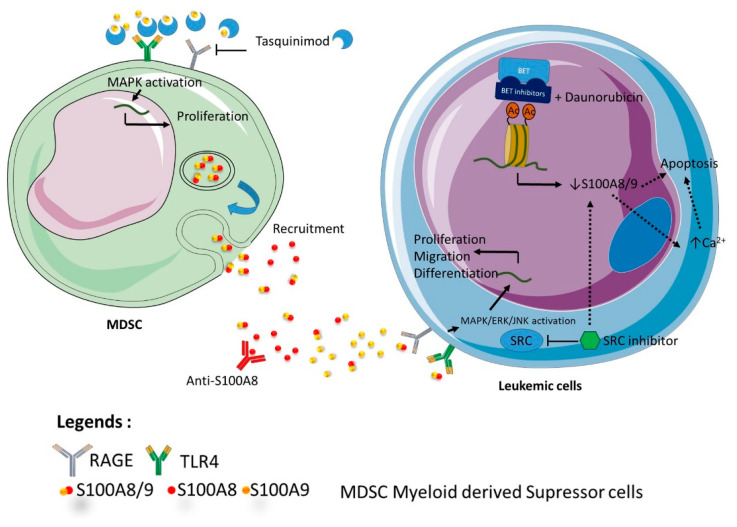
Representation of the different therapeutic strategies targeting S100A8 and/or S100A9. The figure indicates the different ways of targeting S100 A8 and/or S100A9 at different stages of development of acute leukemia.Tasquinimod targets the microenvironment by preventing interactions between TLR4, RAGE and S100A9. Anti S100A8 antibodies induce differentiation, growth arrest and prevent MDSC recruitment. BET inhibitors and daunorubicin induce together apoptosis by S100A8/A9 downregulation. Src inhibitors indirectly downregulate S100A8/A9.

**Table 1 molecules-26-01323-t001:** Landscape of S100A8 and S100A9 therapeutic’ strategies in acute myeloid and lymphoid leukemia. The table shows a synthesis of the main strategies that target S100A8 and/or S100A9 and their impact and mechanisms of action in acute leukemia. The last column indicates on-going clinical trials, including some other cancers.

Mechanism of S100A8 and/or S100a9 Targetin	Impact on Tumor Cells and S100 Protein Expression	Cancer Models	Experimental Models	Others Clinicial Trials	References
**Anti-S100A8 Antibodies**	Induction cell differenciation, growth arrest and prolonged survival	AML	AML mice models	Actually, no trial	[32]
**Src kinase inhibitor PP2**	Indirectly downregulation of S100A8 and S100A9 (but also other S100 proteins i.e., S100A4) Synerziges with glucocorticoid	ALL (*MLL*-rearranged ALL)	ALL cells lines	Actually, no trial	[66]
**Quiniline-3-carboxamides i.e., Tasquinimod**	Direct inhibition of S100A9 Depletion of Myeloid derived suppressor cell though TLR4	Prostate tumors and lymphoma	Lymphoma model	▪ Phase II Metastatic castration prostate Cancer ▪ Myeloma (Phase I)	[74]
**Short hairpin RNA in cancer cells**	Diminution of migration and invasion	Colon and lung carcinoma cells	Mice models liver metastasis	Actually, no trial	[30]
**Bromo and extra Terminal Domain Family (BET) inhibitors (i.e., JQ1)**	Downregulation of S100A8 and S100A9 by JQ1 in some patient cells cultures and in some lines cells dependent on mutational status Not well understood, probably autophagy Synerziges with Daunorubicin	AML	18 patients and acute leukemia cell lines	Numerous phase 1 and 2 studies (myelofibrosis, acute leukemia, lymphoma…)	[76,77]

## Data Availability

The data presented in this study are available on request from the corresponding author.

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
