# Peer review of "Pathogenic Roles of S100A8 and S100A9 Proteins in Acute Myeloid and Lymphoid Leukemia: Clinical and Therapeutic Impacts"

_molecules, 2021, doi:10.3390/molecules26051323_

Round 1

Reviewer 1 Report

This is a comprehensive review on the pathogenic role of S100A8 and S100A9 proteins in acute leukemia. I have some observations concerning how the manuscript might be improved as follows:

1)         The manuscript should be more coincided, focusing mainly on the learning goals of the review that should be better defined

2)         To facilitate the understanding for general readers, I recommend adding a figure explaining interactions between S100A8 and S100A9 and the inflammatory reactions in pathogenesis and progression of tumors.

3)         The important points of the role of S100A8 and S100A9 on the pathogenesis of acute leukemia and the related target therapy on development should be better presented, for example adding an additional figure. 

4)         Could the authors comment on the extent to which the S100A8 and S100A9 proteins might be of particular relevance to hematologic malignancies, characterized by an inflammatory environment, specifically AML with myelodysplasia-related changes or AML secondary to myeloproliferative neoplasm? A comment on the possible correlation of these proteins according to the AML genetic and molecular profile would also be of interest.

5)         It would be of interest if the authors could speculate on what the therapeutic landscape of S100A8/A9 targeting strategy might look like in the coming years. Will there be specific patients with acute leukemia for whom these specific target treatments might be preferred?

6)         Line 127 “AML1 and 2” should be AML M1 and M2

7)         Data on AML with IDH mutation (lines 150-154) should be reported before the description of ALL, independently if these data derived from pediatric patients.

Author Response

Responses to Reviewer-1:

Remark 1: Following the reviewer's recommendation, we slightly changed the organization of the chapters and their name. Moreover, at the end of the introduction, a brief paragraph describes the main objectives of the journal in order to give readers a global view of the points discussed in the article (line 95-102).

Remark 2: To facilitate the overall understanding of the multiple roles of S100A8 and S100A9 in acute leukemia, we propose in this new version of our manuscript an additional figure (Figure-3) representing the different impacts of S100A8 and S100A9 on leukemia cells and their role on the hematopoietic microenvironment.

Remark 3: In order to better identify and visualize the different possibilities of therapeutic targeting, we now show the different strategies targeting S100A8 and/or S100A9 with their mechanisms on a new figure (Figure-4).

Remark 4: Regarding the correlations between S100A8/A9 and the molecular and genetic profile of patients, we fully agree that this is a very important issue. This knowledge would most likely lead to a better understanding of some of the contradictory effects described in the literature from cohorts of patients, who are not sufficiently characterized. Unfortunately, the bibliographical data on these correlations between S100 A8/9 and molecular profile of patients are very low with the exception of data specific to the APL, or re-arranged AL MLL. Hence, it’s difficult to be extensive on this point. Nevertheless, we highlighted this issue and better underscore the known relationship in the new manuscript (line 134-140)

Remark 5: In addition to the elements provided in the new figure-4, the paragraph on the therapeutic value of targeting S100A8 and S100A9 has been reworked. In particular, it now presents a synthesis of therapeutic strategies that could be proposed, in the current state of knowledge, at the different stages of development of acute leukemia (pre-leukemic states, de novo or secondary AL) (lines 409-418).

Remark 6 and 7: The corrections proposed by the reviewer were made.

In addition, in order to improve its quality, the grammar and syntax of the manuscript was review and corrected by paper true (cf certificate in attached piece)

Reviewer 2 Report

Mondat, Chevalier and Mossuz have provided a timely review of the role of calcium binding proteins S100A8 and S100A9, focusing primarily on hematological malignancies, namely AML and ALL. They provide a very good survey of the latest studies investigating the role of S100A8/A9 in these diseases and in processes such as cell survival and drug resistance. While there have been other reviews on S100A8/A9 in other conditions and in cancer in general, this is the first focused review of studies investigating S100A8/A9 in acute leukemia.

Specific comments:

  1. To provide a better overview and help readers to understand the many roles of S100A8/A9 in different cellular processes, particularly cell survival/death, it would be helpful to include a schematic figure.
  2. Page 1, line 30: 1q21 gain/amplification occurs commonly in some cancers, therefore it is not correct to say “1q21 is frequently rearranged in cancer”. In addition, supportive references should be provided.
  3. Page 2, line 87-88: references about the roles of S100A8 and S100A9 in solid tumors are needed.
  4. With regard to the role of S100A8/A9 in the bone marrow microenvironment, the study by De Veirman and colleagues investigating extracellular S100A9 in multiple myeloma bone marrow environment (De Veirman et al., Cancer Immunol Res, 2017) should be included.
  5. The manuscript requires careful review since there are numerous small errors throughout the text. The language also requires careful review.

Example errors:

  • Page 2, line 53, NADPH oxidase, short name NOX should be in parentheses.
  • Page 2, lines 74-75: “(psoriasis, rheumatoid arthritis, chronic inflammatory, intestine diseases…)” should be corrected to “(psoriasis, rheumatoid arthritis, chronic inflammation, intestinal diseases, etc.)” and “(TNF, IL 6…)” should be corrected to “(TNF, IL6, etc.).
  • Page 4, 157-159: AML1, AML2, etc. should be corrected to AML M1, AML M2, etc. Also, “AM” on line 159 should be corrected to “AML”.
  1. Not all labels in the bar graphs presented in Figure 1 are clear. For example, the “H4” and “H24” labels in Figure 1C are not fully visible.
  2. The presentation of Table 1 is not optimal and could be improved.

Author Response

Response to Reviewer 2

Remark 1: To provide a better overview of the many roles of S100A8 and S100A9, as suggested by the reviewer, we provide a new figure drawing the different impacts of both proteins on leukemic cells and on micro-environment (Figure-3). Moreover, in order to better identify and visualize the different possibilities of therapeutic targeting both proteins, we have also incorporated a new figure (Figure-4) showing the related therapy as well as their mechanism of action on leukemic cells.

Remark 2: We made the correction about gain and amplification of 1q21 in cancer and have now provided a supportive figure (Figure-1) that shows the frequency of 1q21 alterations in cancers.

Remark 3: Considering this remark as well as a comment from another reviewer, we have slightly modified the terminal part of the introduction. As this review focuses on acute leukemias, we have decreased the information on solid tumors to include only the broad outlines of the impact of S100A8 and S100A9 in cancers with some informative references. On the other hand, we have added a brief paragraph that outlines the main objectives of the review in order to give readers a better and more precise view of the issues discussed in the article (lines 95–102)

Remark 4: The reference proposed by the reviewer has been added to the bibliography, and it is discussed in the new manuscript (lines 266–69)

Remark 5: The manuscript has been carefully reviewed by the authors. All the errors notified by the reviewer have been corrected. In addition, in order to improve its quality, the grammar and syntax of the manuscript was reviewed and corrected by Paper True (cf certificate in attached piece)

Remark 6 and 7: The Figure-1 is now the Figure-2 in the new manuscript. Figure-2 and Table 1 have been corrected and improved as recommended by the reviewer

Reviewer 3 Report

The authors present a well-organized and thoughtful review of the role S100A8 and S100A9 proteins in acute myeloid and acute lymphoid leukemia. This reviewer recommends publication with the following minor revision. 

After the conclusion there is a hanging figure and table. Figure 1 should be alongside its reference on Pg. 5 in the text. Is figure 1 new data that hasn't been published? If not, where is the reference to this data and does the figure have a copyright? Please incorporate Figure 1 and Table 1 into the text of the article.

Author Response

Reviewer 3

 Regarding the results presented in Figure-2, these are results that have not been published previously and there is no copyright on the figure.

Concerning the remark about the figure, we indicate to the reviewer that this new manuscript includes now 4 figures, with in particular Figure 3 that represents impact of S100A8 and S100A9 on hematopoiesis and leukemic progenitors, and Figure-4 which represents the different therapeutic strategies that aim to target S100A8 and A9

Round 2

Reviewer 1 Report

The authors have addressed the issues I raised in my previous review. The revised manuscript describes now more clearly the pathogenic role of S100A8/A9 proteins in acute leukemias